# Dysregulation of miR-381-3p and miR-23b-3p in skeletal muscle could be a possible estimator of early post-mortem interval in rats

Vanessa Martínez-Rivera[1], Christian A. Cárdenas-Monroy[1],
Oliver Millan-Catalan[2,3], Jessica González-Corona[1], N. Sofia Huerta-Pacheco[4],
Antonio Martínez-Gutiérrez[2], Alexa Villavicencio-Queijeiro[1],
Carlos Pedraza-Lara[5], Alfredo Hidalgo-Miranda[6], María Elena Bravo-Gómez[7],
Carlos Pérez-Plasencia[2,3] and Mariano Guardado-Estrada[1]

[1] Laboratorio de Genética, Ciencia Forense, Facultad de Medicina, Universidad Nacional Autónoma de México, Ciudad de México, México

[2] Unidad de Investigación Biomédica en Cáncer, Laboratorio de Genómica, Instituo Nacional de Cancerologia, Ciudad de México, México

[3] Unidad de Investigación Biomédica en Cáncer, Laboratorio de Genómica, Facultad de Estudios Superiores Iztacala, Universidad Nacional Autónoma de México, Ciudad de México, México

[4] Cátedras CONACYT—Ciencia Forense, Facultad de Medicina, Universidad Nacional Autónoma de México, Ciudad de México, México

[5] Laboratorio de Entomología, Ciencia Forense, Facultad de Medicina, Universidad Nacional Autónoma de México, Ciudad de México, México

[6] Laboratorio de Genómica del Cáncer, Instituto Nacional de Medicina Genómica (INMEGEN), Nacional de Medicina Genomica, Ciudad de México, México

[7] Laboratorio de Toxicología, Ciencia Forense, Facultad de Medicina, Universidad Nacional Autónoma de México, Ciudad de México, México

Corresponding author
Mariano Guardado-Estrada,
mguardado@cienciaforense.facmed.unam.mx

## ABSTRACT

**Background.** The post-mortem interval (PMI) is the time elapsed since the dead of an individual until the body is found, which is relevant for forensic purposes. The miRNAs regulate the expression of some genes; and due to their small size, they can better support degradation, which makes them suitable for forensic analysis. In the present work, we evaluated the gene expression of miR-381-3p, miR-23b-3p, and miR-144-3p in skeletal muscle in a murine model at the early PMI.

**Methods.** We designed a rat model to evaluate the early PMI under controlled conditions. This model consisted in 25 rats divided into five groups of rats, that correspond to the 0, 3, 6, 12 and 24 hours of PMI. The 0 h-PMI was considered as the control group. Muscle samples were taken from each rat to analyze the expression of miR-381-3p, miR-23b-3p, and miR-144-3p by quantitative RT-PCR. The gene expression of each miRNA was expressed as *Fold Change* (FC) and compared among groups. To find the targets of these miRNAs and the pathways where they participate, we performed an in-silico analysis. From the gene targets of miR-381-3p identified in the silico analysis, the *EPC1* gene was selected for gene expression analysis by quantitative RT-PCR in these samples. Also, to evaluate if miR-381-3p could predict the early PMI, a mixed effects model was calculated using its gene expression.

**Results.** An upregulation of miR-381-3p was found at 24 h-PMI compared with the control group of 0 h-PMI and (FC = 1.02 vs. FC = 1.96; $p = 0.0079$). This was the

opposite for miR-23b-3p, which had a down-regulation at 24 h-PMI compared to 0 h-PMI (FC = 1.22 vs. FC = 0.13; $p = 0.0079$). Moreover, the gene expression of miR-381-3p increased throughout the first 24 h of PMI, contrary to miR-23b-3p. The targets of these two miRNAs, participate in biological pathways related to hypoxia, apoptosis, and RNA metabolism. The gene expression of *EPC1* was found downregulated at 3 and 12 h of PMI, whereas it remained unchanged at 6 h and 24 h of PMI. Using a multivariate analysis, it was possible to predict the FC of miR-381-3p of all but 6 h-PMI analyzed PMIs.

**Discussion**. The present results suggest that miR-23b-3p and miR-381-3p participate at the early PMI, probably regulating the expression of some genes related to the autolysis process as *EPC1* gene. Although the miR-381-3p gene expression is a potential estimator of PMI, further studies will be required to obtain better estimates.

## INTRODUCTION

The post-mortem interval (PMI) is defined as the time elapsed between the dead of an individual and the time the body is found; it being relevant for forensic purposes (*Maile et al., 2017*). At early PMI (3–72 h after death), morphological changes appear, such as decay of temperature (*algor mortis*), cadaveric stiffness (*rigors mortis*), and changes in body coloration (*livor mortis*) (*Lee Goff, 2009*; *Maile et al., 2017*). The identification of these morphological changes is helpful to estimate the PMI. In the meantime, a process called autolysis occurs in the cells of a dead body, characterized by an absence of inflammatory response and cell destruction due to liberation of the enzymes of some organelles (*Tomita et al., 2004*).

However, the occurrence of these external morphological changes could vary due to different factors such as the environment, cause of the death, among others; this can make it difficult to estimate PMI (*Madea, 2016*). Thus, other methods have been developed where some components in vitreous humor or synovial fluid are quantified for PMI estimation (*Madea et al., 1994*; *Madea, Kreuser & Banaschak, 2001*; *Zilg et al., 2015*; *Madea, 2016*; *Ansari & Menon, 2017*). Nevertheless, as with physical changes, variations in the quantifications of these elements reduce the confidence in the calculation of PMI (*Muñoz Barús et al., 2002*; *Madea, 2016*).

Other molecules that have been studied for PMI estimation are nucleic acids (*Koppelkamm et al., 2011*; *Itani et al., 2011*). For instance, RNA degradation has been studied in different tissues throughout the PMIs (*Koppelkamm et al., 2011*). Although it is expected that after the death of an individual the RNA transcription halts, there are many studies that analyzed the expression of some genes at different PMIs (*Pozhitkov et al., 2017*). After death, transcriptional activity has been found in several tissues analyzed in humans and other organisms (*Vishnoi & Rani, 2017*; *Pozhitkov et al., 2017*; *Ferreira et al., 2018*). For instance, a study performed in mice and zebra fish found an upregulation of genes that

participate in several biological processes such as stress, immune response and apoptosis, among others (*Pozhitkov et al., 2017*). In humans, the changes of transcriptional activity at early PMI depend on the analyzed tissue, as well as the rate of RNA degradation in them (*Ferreira et al., 2018*). Although there is not a complete understanding of the underlying mechanism of this transcriptional activity at PMI, it is suggested that epigenetic regulation could be involved (*Pozhitkov et al., 2017*).

The miRNAs are small 22 nucleotide-length non-coding RNAs which can post-transcriptionally regulate the expression of genes implicated in several pathways (*Vishnoi & Rani, 2017*). Due to their small size, the miRNAs endure extreme conditions without degradation, making them suitable for forensic purposes (*Wang et al., 2013*; *Lv et al., 2014*). In fact, it has been reported that miRNAs regulate several processes such as apoptosis and inflammation, which are implicated in the process of body decomposition (*Chen et al., 2018b*; *Zhou et al., 2019*; *Jiang et al., 2020*). On the other hand, a continuous expression of some miRNAs at different PMI has been found in the spleen, heart muscle, brain, and bone on both rats and humans (*Li et al., 2014*; *Lv et al., 2014*; *Nagy et al., 2015*; *Na, 2020*).

For this work we analyzed the expression of miR-144-3p, miR-23b-3p, and miR-381-3p, which participate in apoptosis and inflammation, in rat skeletal muscle at early PMI. In fact, it has been reported that miR-144-3p, miR-23b-3p and miR-381-3p regulate the gene expression of BCL6, PROK2, and IL15RA, respectively, which were found to be altered at the PMI (*Pozhitkov et al., 2017*; *Kozomara, Birgaoanu & Griffiths-Jones, 2019*). First, we established a PMI rat model to analyze the expression of these three miRNAs in rat skeletal muscle at different post-mortem intervals. On the other hand, we performed an in-silico analysis to identify the gene targets of these miRNAs to quantify one of them in these samples. Finally, the expression of *EPC1*, which is target of miR-381-3p in these samples, was analyzed.

# MATERIALS & METHODS

## PMI rat model

A total of 25 adult male Wistar rats was selected for the study; all with an average weight of 200 +/−20 gr. These rats were sorted into five groups, which correspond to the 0, 3, 6, 12 and 24 h of post-mortem interval (h-PMI). The 0 h-PMI was considered the control group. The rats from 3, 6, 12 and 24 h-PMI groups, were euthanized by cervical dislocation and placed in a Binder KBW 240$^{TM}$ climatic chamber with a constant temperature of 25 °C. After the PMI time elapsed in each group, the presence of internal and external morphological changes was evaluated on every rat. Every rat was physically explored, in a cephalocaudal fashion, to evaluate the presence or absence of *algor mortis* (AM), *livor mortis* (LM), *rigor mortis* upper body (RMU), *rigor mortis* lower body (RML), drying (DR), generalized edema (ED), hair loss (HL), abdomen green discoloration (AGD), and abdominal distention (AD), which are physical signs present at early post-mortem interval (*Dix, 1999*; *Brooks, 2016*). Also, each animal was dissected to evaluate the presence of brain liquefaction (BL), brain edema (BE), discoloration of liver (DL), loss of liver consistency (LLC), muscle *livor mortis* (MML), bowel swelling (BS), ascites (AS), and loss

of muscle consistency (LMC). Once the evaluation was performed, 200 mg of femoral muscle sample was obtained and stored at −80 °C until analysis. The rats from the control group were euthanized by cervical dislocation, and muscle samples were taken immediately and stored at −80 °C until analysis. As in the other PMI-groups, control group rats were externally and physically evaluated for the presence of cadaveric signs. All procedures were evaluated and approved by the local ethic and scientific committee and the committee for the care and use of laboratory animals (CICUAL) of the Faculty of Medicine from the National Autonomous University of Mexico (UNAM) with approval number 102-2018, and 027-CIC-201, respectively; the procedures were also performed in strict accordance to local (NOM-062-ZOO-1999) and international norms of laboratory animals handling.

## RNA extraction

For miRNAs analysis, total RNA was extracted from the rat skeletal muscle samples using glass beads for rupture and Trizol[TM] Reagent. In brief, a fraction between 50 and 100 mg of frozen tissue was collected in a 2 mL tube with 2 mm glass beads (ZR BashingBead Lysis Tubes, Zymo Research) and 1 mL of Trizol[TM] Reagent. Then, 200 μl of clorophorm was added and mixed to be centrifuged at 12, 000 g for 15 min at 4 °C. After this step, total RNA extraction was performed according to manufacturer's recommendations. The obtained RNA was quantified with an UV- spectrophotomer NanoDrop[TM] 2000 (Thermo Scientific), and the integrity was evaluated qualitatively in agarose gels.

## miRNAs quantification by RT-PCR

From the total RNA of muscle samples, miRNA cDNA was synthetized using the kit TaqMan Advanced miRNA cDNA Synthesis kit (Applied Biosystems). This kit performs the poly(A) tailing reaction via adaptor ligation previous to the miRNA cDNA synthesis. All reactions were performed according to the manufacturer's protocol. Gene expression of miR-144-3p, miR-23b-3p, and miR-381-3p was evaluated by qRT-PCR using the TaqMan® probes rno481325_mir, rno478602_mir, and rno481460_mir, respectively. The miR-361-5p (rno481127_mir) was used as internal control, since it has been seen that its expression is stable under extreme conditions, such as cancer (*Della Bella & Stoddart, 2019*). The miRNas quantification was performed in a StepOne[TM] Real-Time PCR System (Thermo Fisher Scientific, Waltham, Massachusetts, U.S.A) using 10 ng of total cDNA, 0.5 μl of the TaqMan® Advanced miRNA Assay (20X), and 5 μl of TaqMan® Fast Advanced Master Mix (2X) in a total volume of 7.5 μl. Each quantitative RT-PCR was incubated at 95 °C for 20 s, then at 95 °C for 3 min with 40 cycles of denaturation and annealing/extension at 60 °C for 30 s. Each miRNA was analyzed separately, and each sample was run by triplicate. Each miRNA was relatively quantified with the $2^{-\Delta\Delta CT}$ method, and data is presented as fold change (FC), which is the gene expression normalized to miR-361-5p and relative to the 0 h-PMI group (*Livak & Schmittgen, 2001*).

## mRNA quantification by RT-PCR

From the total RNA extracted from the rat's muscle samples, cDNA was synthetized using High-Capacity cDNA Reverse Transcription Kit (Thermo Fisher Scientific, Waltham, Massachusetts, U.S.A). The reaction was performed in a total volume of 20 μl, which

included 1 μg of RNA, 2 μl of 10X RT Buffer, 0.8 μl of 25X dNTP Mix, 2 μl of 10X RT Random Primers, and 1 μl of MultiScribe Reverse Transcriptase. Reactions were incubated at 25 °C for 10 min, at 37 °C for 120 min, and at 85 °C for 5 min; then they were stored at −20 °C for further analyses. Gene expression of *EPC1* (Rn01538512_m1) was evaluated with RT-PCR using TaqMan® probes. To normalize the expression of *EPC1*, the *ACTB* gene (Rn00667869_m1) was used in the analysis as a housekeeping gene. Quantification was performed in a StepOne™ Real-Time PCR System (Thermo Fisher Scientific, Waltham, Massachusetts, U.S.A) using 45 ng of total cDNA, 1 μl of the Taqman™ Probe, and 10 μl of Taqman® Master Mix in a total volume of 20 μl. Each quantitative RT-PCR was incubated at 50 °C for 2 min, then at 95 °C for 10 min with 40 cycles of denaturation at 90 °C, and annealing/extension at 60 °C for 60 s. Each gene was analyzed separately and ran by triplicate in all samples. The average CT threshold calculated for each sample was used to relatively quantify the *EPC1* gene expression using the $2^{-\Delta\Delta CT}$ method expressed as Fold-Change (FC) (*Livak & Schmittgen, 2001*).

## The miRNAs target identification analysis and their pathways

The gene targets of the miRNAs miR-23b-3p, and miR381-3p were identified *in silico* using several bioinformatic databases, that included predicted and experimentally validated targets. For predicted targets, the databases DIANA-microT-CDS (*Reczko et al., 2012*; *Paraskevopoulou et al., 2013*), ElMMo (https://mirz.unibas.ch/ElMMo3/index.php), MicroCosm (http://multimir.ucdenver.edu/), miRanda (http://mirdb.org/), miRDB (http://mirdb.org/), PicTar (https://pictar.mdc-berlin.de/), PITA (Segal Lab of Computational Biology, https://genie.weizmann.ac.il/pubs/mir07/mir07_prediction.html), and TargetScanHuman (http://www.targetscan.org/vert_72/) were used. Additionally, the validated targets were searched in the miRecords (http://c1.accurascience.com/miRecords/), miRTarBase (*Chou et al., 2018*), and TarBase (*Karagkouni et al., 2018*). For each miRNA, we selected only those target genes which were present in at least three or more databases (Script in Supplemental Data). Moreover, to identify the biological pathways where these gene targets participate, we analyzed them further with the Gene Set Enrichment Analysis (GSEA) from WEB-based Gene Set Analysis Toolkit (WebGestalt, http://www.webgestalt.org/). Only those biological pathways with a False Discovery Rate (FDR) less than 0.05 were considered.

## Statistical analysis

The presence or absence of morphological changes through the PMI in rats was evaluated with the Multiple Correspondence Analysis. The FC of each miRNA was compared among the PMIs with the non-parametric Kruskal Wallis and the Mann U Whitney test. Using the Cohen's d calculation and expecting a large effect size ($d = 0.8$), we expected a statistical power of 0.8 with the sample size of each group in the present study (*Lakens, 2013*). The dependence and association of the morphological changes with the PMI were evaluated with the Pearson's Chi-squared test and Cochran-Armitage test, respectively. To explore whether there is an association between the post-mortem interval and the expression of miR-381-3p and miR-23b-3p a Spearman Rho correlation was calculated. A Mixed Effect

Model was calculated considering the FC, the morphological changes of brain liquefaction and cerebral edema with PMI, as an independent variable ($y = X\beta + Z\gamma + \varepsilon$; where: $y$ is the response vector of all the observations; $X$ is a fixed effects design matrix; $\beta$ is a p fixed effects vector; $Z$ is a random effects design matrix; $\gamma$ is a random effects vector; and $\varepsilon$ is the observation error vector) in the "lmerTest" package with the active option of the REML (Restricted Maximum Likelihood). All statistics were performed with the R-project software (https://www.r-project.org/). The dataset and scripts can be found in github (https://github.com/nshuerta-ForenseUNAM/Dysregulation_miRNA).

## RESULTS

### Morphological changes are heterogenous at different post-mortem intervals in rats

Internal and external physical changes were evaluated at different post-mortem intervals in 25 rats. The external physical changes evaluated included *algor mortis* (AM), *livor mortis* (LM), *rigor mortis* upper body (RMU), *rigor mortis* lower body (RML), drying (DR), generalized edema (ED), hair loss (HL), abdomen green discoloration (AGD) and abdominal distention (AD). All but RML, ED and AGD, appeared after the first 3 h of PMI (see Table 1). On the other hand, the AGD was not observed until the 12 h of PMI. In the case of *rigor mortis*, the RMU and the RML was only seen from 3 to 6 h-PMI and 6 to 12 h-PMI, respectively.

All animals were dissected to evaluate the following macroscopic characteristics: brain liquefaction (BL), brain edema (BE), discoloration of liver (DL), loss of liver consistency (LLC), muscle *livor mortis* (MML), bowel swelling (BS), ascites (AS), and loss of muscle consistency (LMC). As in the external characteristics, all these changes but for AS and LMC were gradually seen after 3 h of PMI. Interestingly, AS and LMC did not appear until 24 h-PMI and were present in 60 and 80% of the analyzed animals, respectively.

These morphological changes were evaluated in a Multiple Correspondence Analysis (MCA) in order to see how these characteristics group through the PMI (See Fig. 1A). The MCA plot captured at least the 74.2% of the data and, as it was expected, the morphological characteristics of the 0 h-PMI and the 24 h-PMI were located opposite from each other. Also, there were PMIs that clustered into three groups because they shared some morphological characteristics among them (See Fig. 1B). The first group included 0 and 3 h-PMI, the second the 6 and 12 h-PMI, and the third the 24 h-PMI. Group I is consistent with the few morphological changes present within the first 3 h of PMI, while group III is where all the early PMI physical characteristics have been established. However, in group II, there were characteristics that were present or absent in both, the 6 and 12 h-PMI, which did not allow to differentiate them in the MCA. These data suggest that the estimate of PMI using morphological changes could be more precise between 0 to 3 h-PMI and at 24 h-PMI of death.
**Table 1  Presence of the external and internal macroscopic morphological characteristics in rats at different post-mortem intervals.**

| Morphological changes | Frequency % | | | | |
|---|---|---|---|---|---|
| | 0 h-PMI | 3 h-PMI | 6 h-PMI | 12 h-PMI | 24 h-PMI |
| **External** | | | | | |
| *Algor mortis* (AM) | 0 | 100 | 100 | 100 | 100 |
| *Livor mortis* (LM) | 0 | 100 | 100 | 100 | 100 |
| *Rigor mortis* upper body (RMU) | 0 | 100 | 100 | 0 | 0 |
| *Rigor mortis* lower body (RML) | 0 | 0 | 100 | 100 | 0 |
| Drying (DR) | 0 | 100 | 100 | 100 | 100 |
| Generalized Edema (ED) | 0 | 0 | 100 | 100 | 100 |
| Hair loss (HL) | 0 | 100 | 100 | 100 | 100 |
| Abdomen green discoloration (AGD) | 0 | 0 | 0 | 40 | 100 |
| Abdominal distention (AD) | 0 | 100 | 100 | 100 | 100 |
| **Internal** | | | | | |
| Brain liquefaction (BL) | 0 | 40 | 60 | 100 | 100 |
| Brain edema (BE) | 0 | 20 | 100 | 100 | 100 |
| Discoloration of liver (DL) | 0 | 60 | 80 | 100 | 100 |
| Loss liver consistency (LLC) | 0 | 20 | 60 | 80 | 100 |
| Livor mortis muscle (LMM) | 0 | 60 | 60 | 100 | 100 |
| Bowel swelling (BS) | 0 | 20 | 100 | 100 | 100 |
| Ascites (AS) | 0 | 0 | 0 | 0 | 60 |
| Loss muscle consistency (LMC) | 0 | 0 | 0 | 0 | 80 |

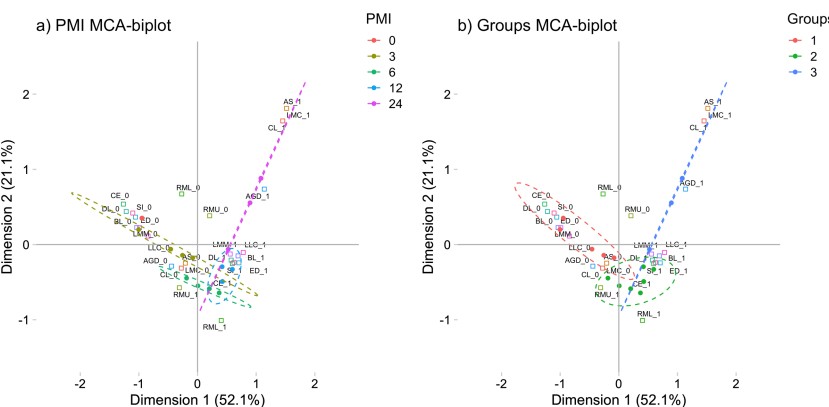

**Figure 1  Multiple correspondence analysis (MCA) between the early post-mortem interval and the presence of morphological changes in rats.** (A) MCA of the analyzed PMIs (0, 3, 6, 12 and 24 h) and the presence of internal and external morphological changes. (B) MCA of the PMIs groups I (0 and 3 h), II (6 and 12 h) and III (24 h), and the presence of internal and external morphological changes. Both plots represented the greatest cumulative variability and could capture at least the 74.2 % of the data. Abbreviations of the morphological characteristics are shown in Table 1. Dots represent each rat, while squares are the dichotomic presence of morphological changes (0: absence, 1: presence). The dashed line ellipses shown the distribution of the rats throughout the PMI.

## miR-381-3p and miR-23b-3p showed gene expression imbalances throughout different PMIs in rats

The gene expression of miR-381-3p, miR-23b-3p, and miR-144-3p was analyzed using qRT-PCR in skeletal muscle of rats exposed to the aforementioned PMIs (see material and methods). Interestingly, miR-381-3p was found upregulated at the 24 h-PMI group of rats compared to the 0 h-PMI control (FC = 1.02 vs. FC = 1.96; $p = 0.0079$, Mann $U$ Whitney test; Fig. 2A). When the FC of miR-381-3p was analyzed at different PMIs, the gene expression of this miRNA had a J-shape curve (see Fig. 2A). First, within the three hours of PMI, the expression of miR-381-3p was downregulated (FC = 0.73), compared to controls, and this difference was statistically significant ($p = 0.0317$, Mann U Whitney test; Fig. 2A). In fact, the difference in the miR-381-3p gene expression was more evident when comparing the 3hr-PMI with the 24 h-PMI ($p = 0.0079$, Mann U Whitney test). Nevertheless, after the 3 h-PMI, the expression of this miRNA gradually increased from 6 h of PMI to 24 h of PMI interval (see Fig. 2A). As it was expected, the difference in the gene expression between 3 h-PMI and 12 h-PMI was statistically significant ($p = 0.032$, Mann $U$ Whitney test).

Contrary to miR-381-3p, the gene expression of miR-23b-3p decreased as the PMI increases to 24 h. The gene expression of miR-23b-3p was downregulated at 24 h-PMI compared to 0 h-PMI, and this difference was statistically significant (FC = 1.22 vs. FC = 0.13; $p = 0.0079$, Mann U Whitney test; Fig. 2B). Interestingly, when the FC of miR-23b-3p was analyzed by the PMIs, the expression of this miRNA decreased from the 3 h-PMI to the 24 h-PMI (see Fig. 2B). There were significant differences comparing 3 h-PMI vs 24 h-PMI ($p = 0.0079$, Mann $U$ Whitney test), 6 h-PMI vs. 24 h-PMI ($p = 0.0079$, Mann U Whitney test) and 12 h-PMI vs. 24 h-PMI ($p = 0.0079$, Mann U Whitney test).

Finally, although the FC of miR-114-3p decreased from 0 h-PMI to 6 h-PMI, these differences were not significant ($p > 0.05$, Mann U Whitney test; see Fig. 2C). In fact, the FC remained unchanged in the following two post-mortem intervals. These results suggest that there is a dysregulation in gene expression of miR-381-3p and miR-23b-3p as the time of post-mortem interval increases in rats.

## Biological process related to miR-381-3p and miR-23b-3p

Using different miRNAs bioinformatic databases (see material and methods), we identified the target genes of miR-381-3p and miR-23b-3p. A total of 2122 and 2076 genes were found to be regulated by miR-381-3p and miR-23b-3p, respectively (Table S1). With each set of genes, a Gene Ontology enrichment analysis was performed to find the main biological processes where they participate (see material and methods). In the case of miR-381-3p, a total of ten biological processes were found to be regulated by this miRNA (see Fig. 3A). Interestingly, some of these biological processes are related to RNA processing as transcription, synthesis and metabolism. Other processes involved with this miRNA, are the positive regulations of gene expression.

On the other hand, ten biological processes were associated with miR-23b-3p, which were different compared with the target genes of miR-381-3p (see Fig. 3B). For instance, the two most enriched biological processes were those related to hypoxia response and
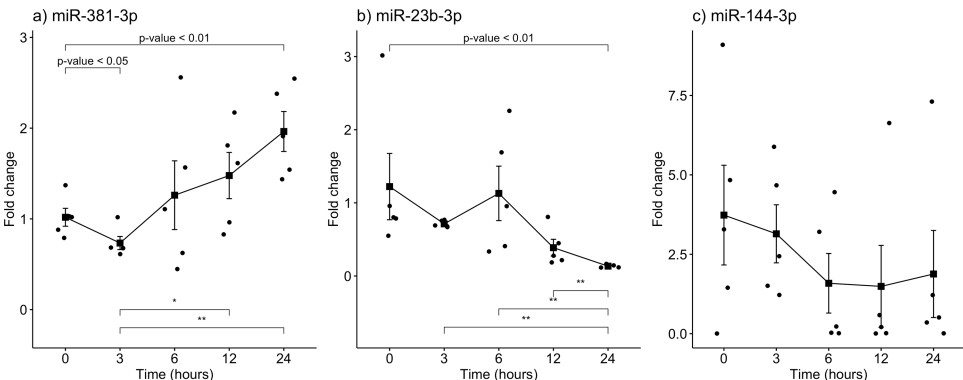

**Figure 2** **Gene expression analysis of miRNAs miR-381-3p, miR-23b-3p and miR-144-3p in rat skeletal muscle throughout the early different post-mortem interval.** The Fold-Change (FC) of miRNAs (A). miR-381-3p (B). miR-23b-3p and (C). miR-144-3p was analyzed in rats skeletal muscle at 0, 3, 6, 12 and 24 hours of PMI using quantitative RT-qPCR. The Fold Change of each miRNA was calculated with the $2^{-\Delta\Delta CT}$ method using miR-361-5p as internal control. The black squares represent the mean of the FC from each group, the whisker corresponds to the 95% confidence interval and the dots are the jittered FC of each sample. Comparisons between the PMI were done with the Mann $U$-Whitney test. * $p$-value $<$ 0.05, ** $p$-value $<$ 0.01.

oxygen levels. On the other hand, there were processes related to the development of the central nervous system. Interestingly, other complex cellular pathways were implicated with this miRNA as positive regulation of signaling, phosphorylation and cell location. Although none of the biological processes where this miRNA participated are related to apoptosis and inflammation, it seems that their function in PMI would be related to the decomposition process of the body.

## Gene expression analysis of *EPC1*

From the target gene list that is regulated by miR-381-3p, *EPC1* was selected for gene expression analyses with qRT-PCR in the same samples used for miRNAs analyses. Despite not seeing a trend as with miR-381-3p, there was a downregulation of EPC1 gene expression at 3 h-PMI (FC = 1.04 vs. FC = 0.58; $p$ = 0.05, Mann $U$ Whitney test) and 12 h-PMI (FC = 1.04 vs. FC = 0.57; $p$ = 0.01, Mann $U$ Whitney test) compared to the control group of 0 h-PMI. These differences were statistically significant (see Fig. 4). Also, there was a slight increase in the expression of *EPC1* gene at 6 h-PMI and at 24 h-PMI compared to 0 h-PMI, though this was not significant. These results indicate that *EPC1* is down regulated or has no change in its expression at different post-mortem intervals.

## Estimation of PMI analyzing gene expression of miR-381-3p

A Spearman Rho correlation was calculated with the FC of miR-381-3p from 3 h-PMI to 24 h-PMI, showing a value of $r = 1$ ($p$ = 0.037). Since a descriptive pattern and association were observed in some variables, we considered that a model could give certainty that effects (fixed or random) affect the fold change of miR-381-3p. To evaluate this, a mixed effect model was calculated considering the FC (dependent variable), the morphological changes, and PMI as independent variables (see material and methods). From the morphological
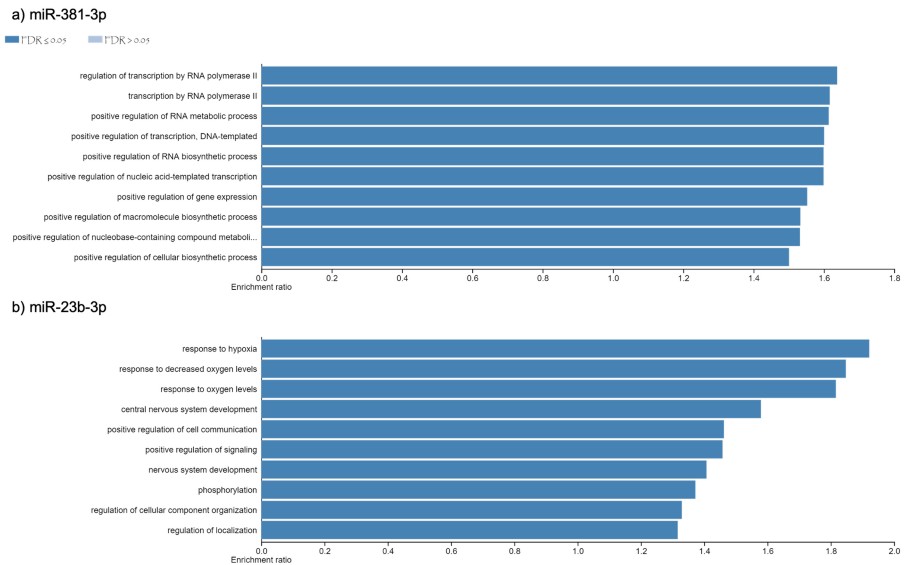

**Figure 3  Gene Ontology enrichment analysis.** The main biological pathways where the target genes of miRNAs (A) miR- 381-3p and (B) miR-23b-3p participate are shown. The $x$-axis correspond to the enrichment ratio and all the biological pathways have a False Discovery rate less than 0.05.

changes analyzed, only the presence of brain liquefaction and brain edema were significantly associated with the FC ($p < 0.01$). An approach to estimate the FC according to PMI of miR-381-3p was done with this model. First, the FC values with respect to change of time (PMI) and the presence or absence of brain liquefaction and brain edema were estimated. Through this model, it is possible to indirectly calculate the PMI, comparing the real FC, with the calculated confidence interval of the estimated FC. The mean FC estimated for 0 h-PMI was 1.01 ±0 (95% CI [1.01–1.01]), 3 h-PMI to 0.73 ±0.04 (95% CI [0.69–0.77]), 6 h-PMI to 1.26 ±0.64 (95% CI [0.62–1.90]), 12 h-PMI to 1.47 ±0 (95% CI [1.47–1.47]), and 24 h-PMI to 1.96 ±0 (95% CI [1.96–1.96]), respectively. It is important that there be no variability in the PMIs of 0, 12 and 24 h, in the estimated values for the FC, so the value in both limits is the same as the mean. According to our results, although the FC of miR-381-3p could be a good predictor of the 0, 3, 12 and 24 h-PMI, the high variability observed at 6 h-PMI hinders the estimation of an accurate interval of PMI according to FC. Albeit the Spearman Rho correlation was negatively significant to the FC of miR-23b-3p according to PMI ($r = -0.9$, $p < 0.05$), there was no significance in the PMI and morphological variables when the mixed effects model was calculated (data not shown).

## DISCUSSION

In the present work we found a gene expression dysregulation of miRNAs miR-381-3p and miR-23b-3p in skeletal muscle tissue of rats exposed to different post-mortem intervals compared to the control group. The miR-23b-3p gene expression decreased from 3 to 24 h of PMI. On the contrary, the gene expression of miR-381-3p increased, with a

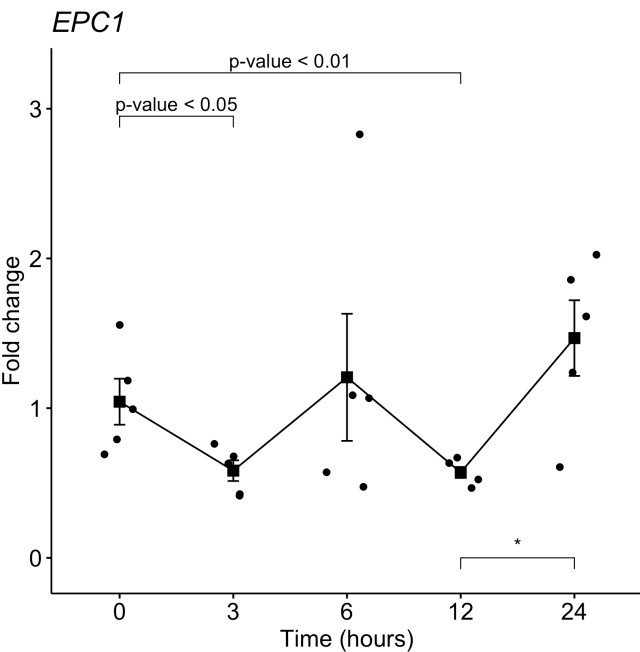

**Figure 4** **Gene expression analysis of *EPC1* gene in rat skeletal muscle at early post-mortem interval.** The Fold-Change (FC) of *EPC1* was analyzed in rats skeletal muscle at 3, 6, 12 and 24 hours of PMI relatively to the 0 h-PMI group, using quantitative RT-qPCR. The Fold Change was calculated with the $2^{-\Delta\Delta CT}$ method using as internal control. The black squares represent the mean of the FC from each group, the whisker corresponds to the 95% confidence interval and the dots are the jittered FC of each sample. Comparisons between the PMI were done with the Mann *U*-Whitney test. * *p*-value < 0.05, ** *p*-value < 0.01.

J-shape curve, as the PMI increased. These two miRNAs regulate the expression of genes which participate in different processes as hypoxia or oxygen depletion sensing, and RNA transcription. Moreover, the gene expression of *EPC1*, which is a gene target of miR-381-3p, was found downregulated or with no change at early PMI, compared to the 0-PMI. Using a mixed effect model, the Fold-change of miR-381-3p could be predicted at 0, 3, 12 and 24 h of PMI.

The presence of several miRNAs in various tissues has been described through the PMI in humans and in rats (*Lv et al., 2017*; *Tu et al., 2019*). However, the analyzed gene expression of some miRNAs has been mainly focused in finding control genes potentially useful for PMI calculation based on gene expression analysis in death bodies. For instance, the gene expression of miR-9 and miR-125b barely fluctuates throughout the different PMI analyzed in spleen (*Lv et al., 2014*). Nevertheless, within the first 24 h of PMI, an upregulation or downregulation of some miRNAs has also been found (*Lv et al., 2014*). In the rat's brain, a slight downregulation of miR-16 was found throughout the 24 h of PMI (*Nagy et al., 2015*). On the contrary, miR-124a, miR-205, and miR-21 were found upregulated within the first 24 h of PMI in brain and skin (*Nagy et al., 2015*; *Ibrahim et al., 2019*). These studies and our results suggest that some miRNAs could be actively involved

in the decomposition process, possibly regulating the expression of other genes, rather than being inert molecules which heavily resist degradation.

After the death of an individual, the autolysis process is seen as a necessary step to achieve body decomposition, and it occurs nearly immediately after the death of the individual (*Zapico, Menéndez & Núñez, 2014*). Nevertheless, more than the liberation of enzymes and proteasomal degradation, the autolysis process is a complex process, where a struggle between survival and pro-apoptotical signals takes place (*Sanoudou et al., 2004*). On the other hand, it has been reported that several genes, some of them related to cell survival, are dysregulated in the PMI, and could last for several days (*Sanoudou et al., 2004*; *Zhu et al., 2017*; *Ferreira et al., 2018*). Thus, it is possible that those genes transcriptionally active favor the body decomposition activating pathways such as apoptosis (*Zapico, Menéndez & Núñez, 2014*). This can also be seen in the biological pathways where the gene targets of the altered miRNAs found in our study participate. Each miRNA regulates different processes that could be related to the autolysis process such as RNA transcription or oxygen levels sensing.

In humans, the miR-381 has been considered as a tumor suppressor in prostate and non-small cell lung cancer inhibiting cell proliferation, invasion and migration through inhibition of nuclear factor-$\kappa$B signaling (*Formosa et al., 2014*; *Huang et al., 2018*). Regarding the PMI, the increase of this miRNA expression could be as a mechanism for promoting apoptosis related to oxidative stress produced by the hypoxia. Another mechanism where miR-381 could participate in PMI is the inflammation inhibition in the autolysis process (*Chen et al., 2018a*). Interestingly, we found a downregulation of EPC1 gene at 3 h-PMI and 12 h-PMI compared to the control group of 0-PMI, and no change at 6 and 24 of PMI. The enhancer of polycomb homolog 1 (*EPC1*) gene codes for a protein member of the polycomb group (PcG) family and is gene target of miR-381 (*Kozomara, Birgaoanu & Griffiths-Jones, 2019*). The coded product of *EPC1* is a part of the NuA4 (Nucleosome Acetyltransferase of H)/TIP60 (Tat Interacting Protein 60) acetyltransferase complex, which participates in several processes to repair DNA double strand breaks (DBSs) and apoptosis (*Zhang et al., 2020*). Also, it has been reported that *EPC1* acts as an oncogene in some types of cancer, such as acute myeloid leukemia (AML), since its suppression triggered apoptosis in cell lines (*Huang et al., 2014*). Our results suggest that one of the mechanisms in of miR-381 that may promote apoptosis could be by down-regulation of *EPC1*, although these results should be confirmed in further works to understand these mechanisms in the PMI.

Contrary to miR-381, the miR-23b-3p showed a gradual reduction of its expression throughout the analyzed PMIs. The miR-23b-3p has been considered as an onco-miR in several types of cancers, such as gastric or breast cancer (*Chen et al., 2012*; *Hu et al., 2017*). Also, it has been found in osteosarcoma that miR-23b-3p promotes cell proliferation, while inhibiting oxidative phosphorylation increasing the lactate levels in these cells (*Zhu, Li & Ma, 2019*). Nonetheless, it is important to emphasize that these results were found in cancer, which could differ from PMI, where the metabolism of the cell is strictly anaerobic (*Donaldson & Lamont, 2015*). The proliferation mediated by miR-23b-3p is due to activation of TGF-$\beta$ signaling by inhibition of *TGIF1* (*Barbollat-Boutrand et al., 2017*).

Also, miR-23b-3p regulates many genes which participate in processes related to oxygen consumption. The participation of this miRNA in the PMI could be in the regulation of the expression of genes related to the response of lower levels of oxygen, which is expected due to the oxygen deprivation in the dead body. For instance, it has been found in mice, that in the first 24 h of PMI, there is an upregulation of hypoxia-related gene transcripts as *Degs2* (*Pozhitkov et al., 2017*). Although we could not discard that the downregulation found on miR-23b-3p at the PMI is due to a higher degradation rate compared to other miRNAs at PMI, its function seems to be closely related with oxygen deprivation present at PMI.

Several works have used the $C_T$ obtained from some genes to estimate the PMI through univariate or multivariate linear regression analyses with high coefficient of determination (*Li et al., 2014*; *Lv et al., 2014*; *Tu et al., 2019*). In our study we used the FC, which relatively estimates the change in expression compared to a control group. From the three miRNAs that we analyzed, the only significant model to estimate the FC was miR-381-3p with a good coefficient of determination ($r^2 = 0.91$). Except for the 6 h-PMI, we were able to estimate the FC according to the 3, 12 and 24 h of PMI. This could be due to a high variability of miR-381-3p expression found at 6 h-PMI, that may be related to individual differences in the autolysis process at this PMI. Interestingly, this variability was also seen in the physical characteristics of the rats at this PMI, which was corroborated in the Multiple Correspondence Analysis. One explanation for this variability is that, at this PMI, there is a heterogeneity in rat's body decomposition; thus, we found rats that presented morphological changes above or below the 6 h mark.

Although we found differences in the gene expression of miR-381-3p and miR-23b-3p in rat skeletal muscle throughout the post-mortem interval, it is important to mention the limitations of the present study. For instance, the sample size from each group in the present study is limited and the findings should be taken as exploratory. Also, it is possible that the analyzed PMIs could not fully reflect the main biological processes occurring in the early post-mortem interval, so further studies involving more PMIs would be required to better define these processes. Since the PMI is a complex biological process, it is probable that there are other miRNAs interacting with other genes in this process. Finally, the conditions of the experiments were performed in an animal model by controlling the environmental conditions, such as temperature and humidity, which can differ drastically from real forensic scenarios. For this reason, we cannot discard that the expression of these miRNAs could vary across different environmental conditions.

## CONCLUSIONS

The gene expression dysregulation of miR-381-3p and miR-23b-3p found in rat muscle at early post-mortem intervals, suggest that these miRNAs participate in the autolysis process. The targets of these miRNAs are involved in pathways related to hypoxia, apoptosis and RNA metabolism. The *EPC1* gene target of miR-381-3p was found downregulated or with no change at an early post-mortem interval. Although miR-381-3p gene expression could be a promising biomarker for post-mortem interval estimation, further studies will be required to refine these predictions.

## ACKNOWLEDGEMENTS

This study was part of the dissertation to obtain the Ms C. degree of Vanessa Martínez Rivera at the Posgrado de Maestría en Ciencias Biológicas, Universidad Nacional Autónoma de México (UNAM).

### Funding

This work was supported by the National Autonomous University of Mexico (UNAM), PAPIIT grant number IA204420. The funders had no role in study design, data collection and analysis, decision to publish, or preparation of the manuscript.

### Grant Disclosures

The following grant information was disclosed by the authors:
National Autonomous University of Mexico (UNAM).
PAPIIT: IA204420.

### Competing Interests

The authors declare there are no competing interests.

### Author Contributions

- Vanessa Martínez-Rivera performed the experiments, analyzed the data, prepared figures and/or tables, authored or reviewed drafts of the paper, and approved the final draft.
- Christian A. Cárdenas-Monroy, Oliver Millan-Catalan and Jessica González-Corona performed the experiments, prepared figures and/or tables, and approved the final draft.
- N. Sofia Huerta-Pacheco and Antonio Martínez-Gutiérrez analyzed the data, prepared figures and/or tables, and approved the final draft.
- Alexa Villavicencio-Queijeiro performed the experiments, authored or reviewed drafts of the paper, and approved the final draft.
- Carlos Pedraza-Lara performed the experiments, prepared figures and/or tables, authored or reviewed drafts of the paper, and approved the final draft.
- Alfredo Hidalgo-Miranda analyzed the data, authored or reviewed drafts of the paper, and approved the final draft.
- María Elena Bravo-Gómez and Carlos Pérez-Plasencia analyzed the data, prepared figures and/or tables, authored or reviewed drafts of the paper, and approved the final draft.
- Mariano Guardado-Estrada conceived and designed the experiments, analyzed the data, prepared figures and/or tables, authored or reviewed drafts of the paper, and approved the final draft.

### Animal Ethics

The following information was supplied relating to ethical approvals (i.e., approving body and any reference numbers):

The committee for the care and use of laboratory animals (CICUAL) of the Faculty of Medicine from the National Autonomous University of Mexico (UNAM) with approval number 027-CIC-2018.

## Ethics

The following information was supplied relating to ethical approvals (i.e., approving body and any reference numbers):

The local ethic and scientific committee of the Faculty of Medicine from the National Autonomous University of Mexico (UNAM) approved this research (approval number 102-2018).

## Data Availability

The script used to select the targets of the analyzed miRNAs and the dataset and scripts used for the miRNAs analysis are available at GitHub: https://github.com/nshuerta-ForenseUNAM/Dysregulation_miRNA.

## Supplemental Information

Supplemental information for this article can be found online at http://dx.doi.org/10.7717/peerj.11102#supplemental-information.

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
