# Peer review of "Dysregulation of miR-381-3p and miR-23b-3p in skeletal muscle could be a possible estimator of early post-mortem interval in rats"

_PeerJ, doi:10.7717/peerj.11102_

## Round 0.1 · original submission · Major Revisions

All reviewers felt this work had merit, but a number of substantive points have arisen during review and in my reading of this work which will require substantive changes to the manuscript and some further experimental work. The reviews are below, and my thoughts are as follows:

I think you need to reconsider how you present the data in this manuscript. The use of fold-change, highlighted by one reviewer, lacks clarity. I would like you to present the data as a mean miRNA expression per time point in each group, at least as a supplemental figure in support of the data shown, but in preference in the actual manuscript itself. You should also describe clearly the mixed effect model using the FC data – it is not clear what you hoped to/have obtained from this analysis.

Please clarify how you identified the target genes of these three miRNAs (see reviewer-2). Having identified these three groups of target genes, two further steps are now required:

(i) clarify for the reader whether these targets regulate similar processes and discuss this point, and
(ii) it is essential that you validate this by measuring levels of some of these targets in your model. Otherwise the data remain speculative and the validity of the model is not established. I regard this as an essential facet of the required alterations of the manuscript.

Regarding the statistical analysis, please describe the Spearman correlation analysis in the methodology, and clarify how the delta ct values were obtained. A power calculation for the use of 25 rats in this study would be a useful addendum to the Materials and Methods. You need to show the extent of changes that can be measured in this experimental size are valid and what their limits are.

All figures need to have careful attention to the size of the text, which is often too small to be readable.
You should spend some time proofreading this manuscript or seek help from a professional service. I have annotated a few examples in the Results section of the enclosed, but there are many more.

I apologise that two of the reviewers of this work are not native English speakers, hence their reviews are a little clumsily worded - if anything isn't clear please email me for clarification via the editorial office.

Reviewer 1 ·

Basic reporting

No comment

Experimental design

No comment

Validity of the findings

No comment

Additional comments

The authors described an interesting work about the correlation of the early PMI in an animal model with the gene expression dysregulation of three miRNAs, miR-381-3p, miR-23b-3p, miR-144-3p. Interestingly, they also evaluated morphological changes of the bodies and correlate those macroscopic characteristics with the molecular information. All sections of the article are well described and the literature citations are appropriate. Even if the results certainly need to be more in-depth and the number of subjects tested should be increased to confirm the association found, this work is a good step forward in understanding the molecular mechanisms at the transcriptional level of the cells that happen in the early stages following death.

Reviewer 2 ·

Basic reporting

Martínez-Rivera et al. performed an interesting study on the expression of miRNAs and morphological patterns at post-mortem in rats.
However, the article needs some improvement before being considered for publication.

The introduction needs more detail; for example, other works involving miRNA expression in the post-mortem should be more explored. It is also not clear the choice of the miRNAs analyzed, and references to them were not mentioned.

The article structure should also be revised. For example, in the materials and methods, the morphological analyzes performed are not described. While that the method of euthanasia of animals is described in lines 130 and 132.

Figure 1 should be improved. The letters are too small.

The abbreviation IPM must be written in full the first time it is cited.

The link "https://github.com/nshuerta-203ForenseUNAM/Dysregulation_miRNA" does not work.

Experimental design

-As previously mentioned, the description of the morphological analyzes and how the descriptive statistics were performed in the study were loss.
-How primers were designed, and their sequences must also be incorporated into the manuscript.
-In the methodology, the authors describe that they used several tools to find the target genes of the analyzed miRNAs (e.g. DIANA, PITA, PICTAR) and selected the target genes that were present in at least three databases (lines 174-184), and identified biological pathways by using GSEA (lines 185-186). However, in lines 271-273, the authors affirmed: "Using the bioinformatic tool WebGestalt (see material and methods), we identified the target genes of miR-381-3p and miR-23b-3p". Please check this out.
-An important point is the concept of fold-change. FC is a measure describing how much a quantity changes between an original and subsequent measurement (e.g., the difference between 0h vs. 3h, or 0h vs. 6h post-mortem). However, the authors provide fold-change values (text and figure 2) for each time-point. Please, give us more information about it. Wouldn't the authors be referring to the mean miRNA expression in each group?
-How was performed the multiple correspondence analysis? Please, provide more details.
-The spearman correlations were not described in the methodology
-The authors performed a mixed effect model, using FC values. What would be the purpose of this analysis? Please insert in the text. In addition, review the use of FC (?) in these analyzes.

Validity of the findings

no comment

Reviewer 3 ·

Basic reporting

This article is interesting, however many questions must be considered.
The 3 microRNA are related with controlling which genes? Apoptotic, necrotic? Because if they are not related, what the fundamental use of this Marker.
Also there are mistakes in expressão the results: I suppose that when they are writing ion expression of gene of miR.... they are expresssing the Gene that this miR regulate?. I did undestand that this is a misexpression. So the entire text must be reviewed. Also misinterpretation in the result I Don't think when increasing or decresing expression of miR is considered downregulation, or upregulation, because miRNA by itself is not regulated. Maybe there is a decrease in formation because the cells are in process of death.
the increase or decrease of miR expression is compared with fresh cell?
I understand that is was used a control miR? I think this strategy is not correct, or was not well written.

Experimental design

It is not clear how value was used to calculated delta delta ct?? The time 0? or internas controls? this data is very important.
In the tissue obtained was added any RNAse inhibitor? It is not mencioned.
Also the term gene expression of miR is not correct.

Validity of the findings

There is no validation. just a experimental Study in 25 mice

Additional comments

There is a lot of misexpression of Basic knowledge in molecular Biology terms
The design must have a more clear expression that can be Made using a workflow .
the methodology in real time must be more clear in the calculation of data
the conclusion is not representative as mentioned

---

## Round 0.2 · accepted · Accept

Thanks for attending to the revisions.

Reviewer 2 ·

Basic reporting

I am satisfied wich how the authors have addressed of my comments. Therefore, I recommend publishing the article "Dysregulation of miR-381-3p and miR-23b-3p in skeletal muscle could be a possible estimator of early post-mortem interval in rats" in PeerJ . I would just like to suggest final revision of the English and text format (e.g. check italic form of genes).

Experimental design

no comment

Validity of the findings

no comment